# Cell and Animal Models for SARS-CoV-2 Research

**DOI:** 10.3390/v14071507

**Published:** 2022-07-09

**Authors:** Eloïne Bestion, Philippe Halfon, Soraya Mezouar, Jean-Louis Mège

**Affiliations:** 1Microbe Evolution Phylogeny Infection, Institut pour la Recherche et le Developpement, Assistance Publique Hopitaux de Marseille, Aix-Marseille University, 13005 Marseille, France; nadeige.bestion@gmail.com (E.B.); phalfon@genosciencepharma.com (P.H.); 2Institue Hospitalo, Universitaire Mediterranée Infection, 13005 Marseille, France; 3Genoscience Pharma, 13005 Marseille, France

**Keywords:** SARS-CoV-2, COVID-19, cell model, animal model, drug

## Abstract

During the last two years following the severe acute respiratory syndrome coronavirus 2 (SARS-CoV-2) pandemic, development of potent antiviral drugs and vaccines has been a global health priority. In this context, the understanding of virus pathophysiology, the identification of associated therapeutic targets, and the screening of potential effective compounds have been indispensable advancements. It was therefore of primary importance to develop experimental models that recapitulate the aspects of the human disease in the best way possible. This article reviews the information concerning available SARS-CoV-2 preclinical models during that time, including cell-based approaches and animal models. We discuss their evolution, their advantages, and drawbacks, as well as their relevance to drug effectiveness evaluation.

## 1. Introduction

In December 2019, the emergence of the coronavirus disease 2019 (COVID-19), caused by the severe acute respiratory syndrome coronavirus 2 (SARS-CoV-2), rapidly evolved into an unexpected pandemic [1]. COVID-19 is a still-expanding global disease involving more than 539,000,000 cases, including over 6,300,000 deaths worldwide [2]. SARS-CoV-2, targets the respiratory tract, focusing on epithelial cells and pneumocytes [3]. Transmission occurs through direct, indirect, or close contact with infected people via saliva, respiratory droplets, or secretions [4]. The SARS-CoV-2 spike (S) protein engages angiotensin-converting enzyme-2 (ACE2) as an entry receptor and is then primed by the transmembrane protease serine-2 (TMPRSS2) [5], two major factors enabling viral entry. Symptoms associated with SARS-CoV-2 broadly include dry cough, fever, headache, and weakness. The respiratory symptoms are remarkably heterogeneous, ranging from almost asymptomatic to acute respiratory distress syndrome that may lead to patient death [6]. A significant portion of patients experience gastrointestinal manifestations, such as diarrhea, vomiting, and loss of appetite [7]. COVID-19 affects the lung, digestive tract, and nervous system and causes multi-organ pathologies associated with immune system dysregulation [8,9]. Clinical reports have further disclosed prolonged and persistent clinical consequences after SARS-CoV-2 infection, including pulmonary fibrosis, thromboembolic events, chest pain, or neurological dysfunctions [10]. The world is thus facing an emergency situation, with medical and societal-associated impacts being increasingly appreciated but still not adequately managed.

To tackle this exceptional situation, a drug-repurposing approach was first attempted before giving way to innovative solutions. In both cases, relevance of the preliminary preclinical findings have been evaluated in more than 6800 clinical trials [11] through the end of November 2021. Trying to curb the COVID-19 health crisis, the clinical translation phase has thus been achieved in record time, shaking up research and medical habits. Many independent and diverse preclinical models have therefore been employed to understand the SARS-CoV-2 infection. In vitro models are compulsory for studying virus biology under highly controlled conditions. They further allow accomplishing substantial drug screening assays. Nevertheless, in vitro models have failed to demonstrate the complexity of the human body and whole cell interactions [12]. In contrast, in vivo models support the elucidation of the COVID-19 pathogenesis, reflecting the diversity of clinical manifestations. These models further elucidate the details of the host-immune response [13]. Extensive preclinical exploration can identify a lead candidate, select the best formulation, and determine the route, duration, and frequency of administration, which aids in supporting the upcoming clinical trial design. To date, disappointing outcomes have been reported regarding COVID-19 trials, as they have been failing to benefit patients [11,14]. An increased failure rate could be attributed to the limitations of time and means associated with the pandemic period. Indeed, several repurposed drug candidates have entered clinical trials even before showing efficacy in animal models [15,16,17]. Some preclinical models were missing SARS-CoV-2 receptors/mechanisms [5], contributing to a partial or inadequate demonstration of the disease and leading to impaired conclusions [18,19,20]. The lack of relevant models and absence of consensus regarding the endpoints monitored [21] reaffirms how important the proper characterization of the preclinical forces deployed is.

In this review, we provide an overview of the different models employed so far for SARS-CoV-2 infection comprehension and therapeutic testing. We discuss their strengths and limitations, especially their involved receptors and tissue origins, as well as their correct recapitulation of human clinical signs. Emerging techniques applied to SARS-CoV-2 research are discussed. We also discuss the importance of the identification of animal reservoirs to guarantee effective disease control. Finally, we outline the indispensable necessity of compiling models to correctly summarize SARS-CoV-2 pathophysiology and to consequently limit clinical trial failures.

## 2. Cell-Based Approaches

### 2.1. Cell Models

#### 2.1.1. Immortalized Cell Lines

Similarly to SARS-CoV [22], the ACE2 receptor was found to be necessary for SARS-CoV-2 entry [5]. A group of cell lines known to express high ACE2 levels were put forward from the outset of the pandemic (Table 1). One such cell line is Vero, which has been broadly used to isolate, propagate, and investigate clinical isolates [23,24,25,26]. This cell line has indeed been proven to be highly susceptible and permissive to SARS-CoV-2 replication [27,28,29]. In contrast to Vero WT cells, the Caco-2 cell line also expresses the SARS-CoV-2 co-receptor TMPRSS2 endogenously, although the TMPRSS2-overexpressing Vero cell model successfully led to an increase in SARS-CoV-2 replication compared to wildtype [30,31].

Vero cells nevertheless do not well-recapitulate the pathophysiological properties of SARS-CoV-2, as they originate from the kidney epithelial cells of an African green monkey [25]. More interestingly, Caco-2 and Calu-3 cells were obtained from human colorectal and lung adenocarcinomas, respectively. They are reported to strongly express both ACE2 and TMPRSS2 receptors [69,70] and to present a strong titer of SARS-CoV-2 particles [13,28] compared to other cells lines originating from the lung and bronchus [29,35,49,71,72] (Table 1). Caco-2 and Calu-3 cells have therefore been widely used and have revealed relevant pathways for SARS-CoV-2 infection [29,37]. The Calu-3 cells are additionally able to polarize, potentially explaining their enhanced susceptibility to SARS-CoV-2 infection [73,74].

It has been found that 2D cell-based models are relevant in understanding the physiopathology and are suitable for high-throughput screening of drugs against SARS-CoV-2. The most frequently reported criterion to quantify the in vitro antiviral potency of compounds is the EC_50_; i.e., the drug concentration in cell culture media that provides 50% maximal antiviral activity. It should be noted that many experimental factors can affect the estimation of EC_50_, including viral strain, multiplicity of infection, drug duration, serum percentage, and cell line model chosen. Chloroquine (CQ), an anti-malarial drug, and its derivative hydroxychloroquine (HCQ) were the first drugs evaluated for their efficacy against SARS-CoV-2 infection. Comparison of the various preclinical studies and clinical outcomes rapidly spotlighted the importance of the features of each investigated model. CQ and HCQ first obtained very attractive EC_50_ results in Vero cells and thus were widely adopted to treat COVID-19 patients [75,76,77]. They were registered in 90 clinical trials worldwide during 2020, sometimes in combination with other drugs [78,79]. However, in the beginning of summer 2020, Hoffmann et al. were the first to demonstrate that CQ does not inhibit SARS-CoV-2 infection of human lung cells [32]. They revealed that CQ efficiently blocked SARS-CoV-2 infection in Vero cells [76,80,81], but not in Calu-3 and Vero/TMPRSS2^+^ cells. CQ inhibition of viral entry was inefficient and absent, suggesting that CQ is unhelpful in ACE2^+^/TMPRSS2^+^ models, and therefore highlighting the importance of the selected model in evaluating compounds of interest. This was followed by preclinical evidence that did not support broad use of HCQ in COVID-19 [82,83].

Cell line models provided valuable insights regarding virus replication kinetics and associated cellular tropism and interactors. A limitation to the use of immortalized cell lines is that they might not entirely reflect the SARS-CoV-2 infectious mechanism and infection rate, lacking population heterogeneity and for some, polarization. Their proliferation rate may also affect virus transfection efficiency. It is also well-established that most of the available cell lines display common mutations including TP53, KRAS, or EGFR, considered to regulate SARS-CoV replication [84,85,86]. Cell lines are nonetheless an essential study model in the SARS-CoV-2 compound screening step, with high throughput and reproducible results.

#### 2.1.2. Primary Cells

Concerning pathogens that induce respiratory symptoms, primary cells originating from human airway epithelia (HAE) have historically been exploited for coronavirus isolation and cultivation [87,88], as well as to examine their tropism. Indeed, upon HAE cultivation at an airway-liquid interface (HAE-ALI), cells differentiated on the porous support exhibiting basal and apical sides. The tissue created is composed of basal, goblet, and ciliated cells, as in physiological conditions [89]. Hence, Hou et al. were able to demonstrate that human nasal/bronchial/alveolar epithelial cultures are permissive to SARS-CoV-2 infection [90]. The investigation of SARS-CoV-2 titers and ACE2 receptor expression from each cell type moreover revealed that the upper respiratory tract is more susceptible to infection than the lower one. Zhu et al. additionally showed that both ciliated and secretory cells from HAE are permissive to SARS-CoV-2 [91]. They also observed that the virus caused a plaque-like cytopathic effect in HAE cultures associated with apoptosis, destruction of epithelium integrity, and beaded changes, as well as lack of ciliary movement. Moreover, the SARS-CoV-2 infection in this model was long-lasting, with recurrent replication peaks [54].

The characterization of the antiviral activity of lead compounds on primary cells remains crucial and highly informative compared to immortalized cell line. Pruijssers et al. showed that remdesivir strongly inhibits SARS-CoV-2 replication in HAE cultures without causing cytotoxicity [92]. In contrast, favorable antiviral results with HCQ in Vero cells were invalidated in the HAE model cultured in an ALI [93]. The study additionally demonstrated that HCQ did not protect the integrity of epithelial tissue during infection. Since such primary cell cultures reproduce major structural features of human respiratory epithelia, drug screening outcomes are more representative.

Another limitation of the model is the absence of immune cells. Considering host-pathogen interactions and immune response following SARS-CoV-2 infection, standard and co-culture systems with induced pluripotent stem cell (iPSC)-derived cells seemed an appropriate strategy [94,95]. Interestingly, studies on the iPSC-derived human lung epithelial system showed that SARS-CoV-2 infection and propagation lead to transcriptional alterations in infected cells [96,97], characterized by a shift to an inflammatory phenotype and changes in cell function [98]. Indeed, the transcriptome of iPSC-derived cells is comparable to their respective primary homologs, allowing them to respond to immune stimuli in a similar way [84]. Histocompatibility issues occurring when examining human immune cells with other cell types were tackled by Duan et al. while co-culturing lung and macrophage-differentiated PSC [99]. Their modeling of lung cell and macrophage crosstalk during SARS-CoV-2 infection revealed synergistic effects of anti-inflammatory macrophages with ACE2 inhibition [99]. The good modeling of cell-intrinsic responses to the immune response caused by SARS-CoV-2 propels iPSC-derived cells as a useful model for drug screening repurposing strategy.

Taken together, primary cell models appear to be essential to validate previous obtained results and confirm potential drug candidates. Despite their more realistic host-virus interactions, primary cells are often limited by the availability of donor materials. It is further evident that primary cells have a limited proliferation capacity, which is a major obstacle to large drug screening campaigns [100]. Another important caveat regarding human lineages derived in vitro from iPSCs is their immature or fetal phenotypes, possibly confounding disease modeling [97]. However, since studies revealed that genetic defects of type I immunity drive COVID-19 severity [101,102], iPSC obtained from a COVID-19 patient tissue biopsy provide useful new tools in developing or testing specific drugs with more personalized-therapy strategies.

### 2.2. 3D Models

Interestingly, 3D-cell models like organoids are ex vivo tissue constructs derived from primary tissues and embryonic stem cells (ESCs) or iPSCs, and exhibit multiple cell types once differentiated [103,104]. They are self-organized assemblies that mimic complex environmental interactions, tissue structures, and functions of the investigated organs [105]. As SARS-CoV-2 infection not only targets the lungs [106], but also the kidney [107], liver [108], and the cardiovascular systems [109], with almost 25% of patients exhibiting gastrointestinal illness [59], organoid modeling may be a useful model in this context. Interestingly, RNA-seq data demonstrated spontaneous expressions of ACE2 and TMPRSS2 in lung [110], colon [111], kidney [112,113], liver ductal [114], and brain organoids [115]. Combined with these observations, bronchioalveolar [116], distal lung [60], kidney, blood vessel [61], small intestinal [62,63], and even liver ductal organoids [114] were reported to be highly permissive to SARS-CoV-2 infection. The use of this model has been of great interest in better understanding the pathophysiology of SARS-CoV-2 infection in human tissues from different locations.

Organoid models provided understanding of the neurological impairment associated with SARS-CoV-2 infection. Ramani et al. reported that SARS-CoV-2 is able to target human neurons [117]. They observed an altered distribution of the Tau protein from axons to soma, hyperphosphorylation, and apparent neuronal death, revealing the potential neurotoxic effect of SARS-CoV-2, including Alzheimer’s disease.Makovoz et al. generated human eye organoids and found that ocular cells express both ACE2 and TMPRSS2 receptors, and are susceptible to SARS-CoV-2 infection [118], contesting previous observations [119,120]. They also pointed out viral tropism, as higher viral titers were observed in the border of the cornea compared to the central corneal zone.A crucial question was raised by Triana et al., who observed no correlation between ACE2 expression level and the copy number of the SARS-CoV-2 genome in intestinal organoids [121]. They additionally revealed that ACE2 expression is downregulated upon SARS-CoV-2 infection. This consequently questions using ACE2 expression as the only key element for anticipating the infectability of cell types.The organoid model may also allow partially investigating ancillary pathways of SARS-CoV-2 infection, such as inflammation and immunity. Even though the resident immune cell activity of the mucosal/epithelial tissue can be monitored, most organoid models lack relevant immune cells; e.g., macrophages and B and T cells that modulate severe disease [122,123]. Focusing on HAE and airway organoid combined models, a robust induction of chemokines similar to what is seen in COVID-19 patients was observed [36,59]. A next important step would be to complexify organoid models using lung tissues from SARS-CoV-2 infected patients and adding their own key immune cells to investigate different forms and phases of the disease.

The use of organoid models to screen drugs has been little explored. Chen et al. performed drug screening assays covering around 15,000 compounds and demonstrated that only a handful of treatments efficiently inhibited SARS-CoV-2 entry in lung organoids [59]. Among the investigated compounds, they highlighted imatinib, which was supported in a clinical trial [124], while results obtained with Caco-2 model did not promote this hypothesis [125]. To the same extent, intestinal organoid models confirmed the absence of benefit of HCQ treatment in SARS-CoV-2 infection [126]. Organoids have become a pertinent and reliable tool to invalidate/promote potential drug candidates before moving to animal models or clinical trials. Patient-derived organoids also enable screening of efficient therapeutic agents within the framework of personalized medicine, especially with cancer patient-derived organoids [127]. As genetic predisposition could be a risk factor in developing severe COVID-19 [101,102], the genesis of mutant organoids for genes previously identified as implicated in coronavirus biology could be of interest to better understand the pathophysiology of SARS-CoV-2 infection. More representative 3D-structured models and personalized medicine capability are organoid features essential to consider regarding the certification of the antiviral activity of a compound. 

Organoids nevertheless lack a vasculature system [104]. Organ-on-chip (OOC) technology is a physiological organ biomimetic system built on a 3D microfluidic cell culture chip. The structure simulates tissue interfaces, mechanical stimulation, and fluid exchanges, as in living organs. Si et al. developed a model including microfluidic bronchial-airway-on-a-chip lined by highly differentiated human bronchial-airway epithelium and pulmonary endothelium [64]. The authors set up a model of SARS-CoV-2 infection with strain-dependent virulence, cytokine production, and the recruitment of immune cells. Others have reproduced key features of the alveolar-capillary barrier based on a human alveolar chip through the co-culture of human alveolar epithelium, microvascular endothelium, and immune cells under fluidic flow [128]. This model revealed that human alveolar epithelial cells were more permissive to SARS-CoV-2 infection than microvascular endothelial cells and provides a new device to screen antiviral drugs. Si et al. used these airway chips containing human lung epithelial primary cells from healthy donors that express high levels of ACE2 and TMPRSS2 to investigate the inhibitory activities of clinically approved drugs against SARS-CoV-2 [66]. They reported that when drugs are administered under flow similar to the maximal ones reported in clinical studies, few drugs significantly inhibited viral entry. However, OOC technology suffers from relatively low throughput [129,130].

3D technology originating from primary cells has emerged as a powerful device. Major benefits are the full range of differentiated cell types, as are present in the target organ, and self-organization, leading to close mimesis of functional and architectural features of corresponding tissues. Organoids also reduce experimental complexity compared to animal models and are further compliant with real-time and kinetic techniques. They could consequently become a new standard for SARS-CoV-2 investigations, as they are more informative than in vitro models and a potent alternative to animal models. However, organ 3D modeling remains complex, with a quite high failure rate.

## 3. SARS-CoV-2 Infection in Animals

### 3.1. SARS-CoV-2 Animal Reservoirs

#### 3.1.1. Wildlife

SARS-CoV-2 isolated from horseshoe bats [30,64] and Malayan pangolins [66,128] exhibited high sequence similarities compared to the whole-SARS-CoV-2 genome. Bats and pangolins were therefore initially considered as a possible source and intermediate hosts for SARS-CoV2, respectively. Interestingly, fruit bats were found susceptible to SARS-CoV-2 infection using intranasal inoculation, presenting mild clinical signs including rhinitis with congener transmission [131]. White-tailed deer express ACE2 that shares a high degree of similarity with humans, and they are highly susceptible to SARS-CoV-2 infection [132,133]. Although they do not present overt clinical symptoms following SARS-CoV-2 infection, they can transmit the virus through direct contact or in vertical transmission from doe to fetus. Griffin et al. raised the question of a plausible risk of SARS-CoV-2 reverse zoonosis through deer mice [134]. Interestingly, they indicated that deer mouse ACE2 differed from human (h)ACE2 by four amino acid residues known to confer efficient binding to the SARS-CoV-2 S protein. These differences were unlikely to have a prejudicial effect on S binding efficiency. The viral infection was persistent for up to 21 days, and the animals exhibited asymptomatic-to-mild disease, with lesions limited to mild lung pathology with elevated levels of inflammatory cytokines. Fagre et al. further described neurological manifestations in the olfactory bulb and tongue that could affect smell and taste senses, respectively, as in humans [135]. Infected animals were also capable of transmitting SARS-CoV-2 to naïve co-housed congeners through direct contact. SARS-CoV-2 zoo animal infections were first noticed with a Malayan tiger, which spread the disease to congeners and lions in the same zoo [136,137]. Independently, puma [138] and snow leopards [139] were declared infected. Considering the susceptibility of the Felidae family, SARS-CoV-2 transmission to the wild animal population should be greater, as predicted.

#### 3.1.2. Domestic Animals

Cats are highly susceptible to SARS-CoV-2 infection and exhibit asymptomatic-to-moderate symptoms, notably interstitial pneumonia, tracheal and nasal inflammation [106], and even severe respiratory clinical signs [140]. Viral transmission to naive congeners has also been monitored [141,142]. Interestingly, human-to-feline transmission has been reported [143], while cat-to-human transmission has not yet been observed. However, since SARS-CoV-2 infected cats shed high levels of virus and transmission to congeners via droplets is possible, the risk of cat-to-human transmission cannot be ignored. In contrast, dogs have low susceptibility to SARS-CoV-2, probably due to the relatively low ACE2 receptor expression in their respiratory tract [144]. Antibody responses have been detected in infected dogs, and virus sequencing suggested that it was the result of human-to-animal transmission [145]. Although current evidence suggests a low risk for domestic pets transmitting SARS-CoV-2 to humans, their close link to their owners encouraged us to consider them as virus carriers and potential intermediate hosts.

#### 3.1.3. Farm Animals

Pigs, chickens, and ducks have been found to have low or even no susceptibility to SARS-CoV-2 intranasal inoculation [131,146]. While domestic pigs are not a suitable model for COVID-19, significant similarities with humans in terms of anatomy, genetics, and physiology promoted them as a useful model for assessing seroconversion and immunogenicity of the upcoming vaccine candidates. Interestingly, following parental inoculation, seroconversion and the presence of neutralizing antibodies were observed in animals, while none resulted in productive infection [147]. The safety and immunogenicity of AVX/COVID-12-HEXAPRO, a viral vector vaccine, were established using a pig model [148]. In contrast, the mink has been reported to experience the first and only known farm anthropozoonotic outbreak to date [149,150]. Among the 16 Netherlands farms investigated, 49% of workers were PCR-positive due to the breeding conditions and mink natural susceptibility to SARS-CoV-2 infection as initially introduced by humans [150]. Three-dimensional structures of the protein region involved in direct interactions with the SARS-CoV-2 S receptor-binding domain (RBD) of the mink ACE2 was found to be similar to humans [151]. Viral variants with mutations on the S protein were reported from infected mink [152]. Interestingly, SARS-CoV-2 infected minks experienced asymptomatic-to-severe symptoms and even death [153]. The viral load obtained from swabs collected *post-mortem* demonstrated higher rates in the throat than in the rectal swabs [13]. As minks might become a continual source of human infection, with alarming SARS-CoV-2 variants, this resulted in mass mink culling worldwide. Although almost all previously cited animals cannot be used as standard experimental models, their investigations brought new insights regarding human-animal transmission and monitoring of an unexpected potential permanent reservoir for SARS-CoV-2. Given their susceptibility and the ability of SARS-CoV-2 de novo mutation, two-way zoonotic transmission must not be neglected, as it might provoke the appearance of new or the reappearance of endemic foci due to animal migrations and human travel.

### 3.2. Animal Models for SARS-CoV-2 Research

In the context of SARS-CoV-2 infection, mice, hamsters, ferrets, and non-human primates (NHPs) have been prominent models (Table 2). Interspecies-related differences are essential aspects to consider, such as host specificity, clinical symptoms, or immune response divergence to evaluate viral pathophysiology.

#### 3.2.1. Mice

Mouse models are currently used to investigate viral pathogenesis and to evaluate both drugs and vaccine responses. As observed for MERS-CoV and SARS-CoV-1 [187,188,189], viral replication was insufficient in wild-type mice using diverse viral strains [28,154,190], due to structural differences in the mouse ACE2 compared to human [191]. Generated in 2007 [191,192], hACE2 transgenic mice were rapidly used during the SARS-CoV-2 pandemic, showing replication of the virus in the lungs leading to local immune cell infiltrates, cytokine storm, and interstitial pneumonia [157]. In contrast, infected K18-hACE2 mice to present weight loss, respiratory distress, and reduced activity, similar to the symptoms observed in humans [157,158,193]. K18-hACE2 [155,157,184] and Hfh4-hACE2 [155] mice models demonstrate encephalitis-related lethality due to viral dissemination in the brain. Viral vector-mediated delivery systems responded to this problem. The AAV-hACE2 model using an adeno-associated virus (AAV) was indeed proposed as a viable model without encephalitis issues, characterized by viral replication until immune clearance, avoiding permanent genetic modification. However, AAV delivery of hACE2 is unspecific, leading to artificial ACE2 expression in non-relevant mouse cell types and generating mouse-to-mouse variation [157,193].

CRISPR/Cas9 knock-in (KI) technology has been used to engineer an hACE2 model [161] that exhibited high viral loads in the lungs, trachea, and brain with no fatalities. Tissue distribution of hACE2 matches clinical signs from COVID-19 patients, making the model highly valuable, contrary to the AAV-hACE2 model. Pathological features of SARS-CoV-2-induced ARDS in humans, including hemorrhage, thrombi, and hyaline membrane-like changes, were however rarely observed in models challenged with nasal inoculation. hACE2-KI mice with intratracheal inoculation of SARS-CoV-2 developed severe form of lung injury [194]. Double hACE2/hTMPRSS2 models were proposed to better recapitulate the precise role of the two proteins during SARS-CoV-2 infection and to better investigate antibody-based therapies, as they will closer resume human-like antibody clearance [195,196,197].

Another way to optimize the mouse model is to use a modified SARS-CoV-2 S protein to enhance binding affinity to the ACE2 mouse [198]. A recombinant strain replicates in the respiratory tract but was cleared within 4-days post-infection, causing mild-to-moderate pneumonia [190]. Leist et al. obtained the SARS-CoV-2 MA10 strain via several passages in mice lungs, in which mutations emerged concerning the S protein, as well as nonstructural proteins and an open reading frame [199]. MA10 infection led to acutely impaired lung function, high levels of proinflammatory cytokines, and non-lethal disease. A mouse-adapted strain, WBP-1, produced lethal lung infections with substitutions increasing the binding affinity to the ACE2 mouse [200]. It has also been suggested to use reverse genetics to reshape the viral RBD, sensitizing mice for infection since the ACE2 mouse protein can be correctly recognized [190]. One caveat of this method is the appearance of genetic mutations that will affect the RB, the primary target site for antibody response.

#### 3.2.2. Hamsters

The Syrian hamster model has already been employed to understand SARS-CoV-1 infection and to test antiviral drugs [201,202,203]. With only 4 amino acids of the critical ACE2 domain differing from the human domain, no genetic manipulations are required for SARS-CoV-2 investigations. This model presents an efficient replication rate in the lungs, with mild-to-severe lung pathological lesions following intranasal virus infection [162,163]. Analysis of lung tissues demonstrated that the organs developed pulmonary inflammation, edema, and cell death [164]. Animals further suffer from weight loss, respiratory distress, and signs of morbidity, such as hunched posture, ruffled fur, and lethargy [164,204]. With a clinical picture similar to that observed in humans, this model has led to a better understanding of the pathophysiological mechanisms (cellular and molecular) of SARS-CoV-2 infection. (1) SARS-CoV-2 infected hamsters exhibited a strong neutralizing antibody protective response to consecutive virus rechallenge [162]. (2) Aged male hamsters developed more severe infection compared to young female hamsters [205]. Moreover, histopathological analysis revealed massive immune cell infiltration into the lungs of young animals compared to older ones. This exuberant innate immune response was linked to the STAT2 signaling that drives viral dissemination in the body, as STAT2-knockout individuals presented lower immune cell infiltration and decreased lung pathology [204]. (3) Airborne and fomite SARS-CoV-2 transmission was observed using the Syrian model. Viral transmission was found to be reduced by the use of surgical mask partitions between cages, limiting respiratory/airborne droplet exchanges and providing the first in vivo experimental evidence to support the potential benefit of wearing masks [206]. More recently, a study explored the different routes of exposure; i.e., contact/fomite/airborne transmission, and found that hamsters presented distinct disease manifestations according to exposure method [207]. Intranasal and aerosol inoculation led to severe respiratory pathology, with an altered shedding profile, while fomite exposure had a delayed shedding pattern associated with milder disease manifestation. Using Syrian hamsters, the Alpha variant was demonstrated to replicate and shed more efficiently in the nasal cavity than other variants, even with a low dose and short duration of exposure [208]. The model additionally showed that a synthetic nanobody targeting RBD [209] and a SARS-CoV-2 neutralizing nanobody [210] protect hamsters from SARS-CoV-2 infection, while favipiravir, a small-molecule drug, was associated with significant toxicity [211]. Recently, the orally or intranasally-delivered Adenovirus type (Ad) 5-vectored SARS-CoV-2 vaccine candidate was demonstrated to have a robust and cross-reactive antibody responses in Syrian hamster model [212]. Animals experienced less lung pathology and airborne transmission compared to control group SARS-CoV-2 challenge. Regarding Omicron variant, Zou et al. showed that prophylactic and therapeutic vaccine-induced ZCB11 administration protects from lung infection in golden Syrian hamsters. Interestingly, ZCB11 also protected animals against the circulating Omicron BA.1 and Delta variants [213].

Bertzbach et al. established Chinese hamsters as a suitable supplemental model for SARS-CoV-2 study, showing pronounced clinical symptoms. The animals indeed present robust viral replication in the upper and lower respiratory tract along with significant weight loss, bronchitis, and prolonged pneumonia associated with acute alveolar damage [214]. The model can spontaneously develop diabetes, making them attractive for investigating COVID-19 with preexisting diabetes [215].

The SARS-CoV-2 infected Roborovski dwarf hamster had a rapid and consistent onset of fulminant clinical disease, including a sudden fall in body temperature, severe acute diffuse alveolar damage, and hyaline microthrombi in the lungs [165]. The authors thus introduced the hamster as a highly susceptible COVID-19 model, having stable and fulminant clinical signs similar to changes described in patients who died from SARS-CoV-2 infection. Such a severe form of COVID-19 has not been reproduced in any experimentally infected animal, positioning this model as a valuable model for screening therapeutics and vaccine candidates in highly susceptible individuals. The therapeutic benefit of molnupiravir in dwarf hamsters infected with omicron variant was evaluated [216]. Interestingly, the authors described a significant virus load reduction in treated males but not in females, highlighting a sex effect regarding the efficacy of the antiviral.

#### 3.2.3. Ferrets

Ferrets have been extensively used for diverse viral pathogenesis. Anatomical and physical ferret characteristics explain why they are generally ideal animals for understanding respiratory virus pathophysiology and mode of action. Ferrets possess an unusually long trachea, permitting easy compartmentation of the upper and lower respiratory tracts [217]. Thus, they are widely employed for studying the transmissibility and tropism of respiratory viruses [218,219,220]. Ferrets have also been employed to study SARS-CoV-1 permissivity and transmission [166,221]. The predominance of alpha 2,6-linked sialic acids in the upper airway epithelia [222,223,224] and the ferret ACE2 critical residues needed for binding to SARS-CoV-2 RBD [225] allow SARS virus attachment patterns to respiratory tissues in a manner similar to humans [226,227]. These characteristics favor high susceptibility to SARS-CoV-2 infection, predominantly in the upper-respiratory-tract location [146,228]. Indeed, after intratracheal virus inoculation, viral RNA was detected in the nasal turbinate, soft palate, and tonsils, but not in the lungs. SARS-CoV-2 vRNA was also detected in rectal swabs, but in lower numbers than nasal samples. Clinical manifestations in ferrets are mainly undetectable-to-mild, including wheezing, lethargy, sneezing, and elevated body temperature [122,229], with acute bronchiolitis in infected lungs for some individuals [167]. Nevertheless, no significant mortality was observed in the ferret model, nor were the severe clinical signs and symptoms observed in humans [230]. Because ferrets can cough and sneeze, they are able to efficiently transmit the virus to their naive cage-mates, indicating that this model is also appropriate for transmission studies [168]. The Kutter et al. study demonstrated efficient transmission through the air between ferrets over more than one meter distance, without discriminating between transmission via small aerosols, large droplets and fomites [228]. Ferrets are further used to assess the immune response to SARS-CoV-2 intranasal inoculation. A similar immune response was observed in lung samples from post-mortem COVID-19 patients compared to ferret samples; i.e., induced high levels of chemokines such as CCL2/8/11 and also a cellular response to IFN-γ [36]. SARS-CoV-2 specific IFN-γ responses were observed in high and medium dose ferrets using lung mononucleocyte samples [230]. Interestingly, it was found that infiltrating macrophages differentiate into M1 or M2 macrophages after SARS-CoV-2 infection [231]. As the ferret immune system reactions are similar to human reactions, they constitute a potentially relevant animal model for therapeutic and vaccine assays. Antiviral efficacies of FDA-approved drugs or under-development compounds have been tested against SARS-CoV-2 infection in ferrets [227,229], supporting the launch of clinical trials.

#### 3.2.4. Chinese Tree Shrew

The Chinese tree shrew is a model which has been successfully employed in replacing primates in biomedical research and drug testing during hepatitis research [232]. Tree shrews exhibit short reproductive cycles and have the advantage of small size. Two studies reported alternative results regarding the use of the mammal in SARS-CoV-2 research. The first study demonstrated undetectable viral RNA in throat and nasal swabs as well in serum samples, while the lung lobes had higher numbers of viral RNA copies [233]. The animals further showed thickened alveolar septa and interstitial hemorrhage with increased levels of aspartate aminotransferase and blood urea nitrogen. The second study found detectable SARS-CoV-2 RNA in nasal, throat, and anal swabs, along with the observation of absent clinical signs except for increased body temperature in females, while lung damage was similar to the previous study [234]. Interestingly, the highest level of virus shedding was observed in young individuals, suggesting their greater susceptibility. The SARS-CoV-2 thus had constrained viral replication and shedding and disparate pathogenesis in tree shrews.

#### 3.2.5. Non-Human Primates

NHPs are thought to be the animal gold standard for modeling diseases, having strong physical and genetic similarities to humans. They have been widely employed for studying viral infection, including the AIDS, Zika, MERS-CoV, and SARS-CoV-1/-2 [235,236,237]. To date, different models have been studied for COVID-19 research, including rhesus, cynomolgus, African green monkeys (AGMs), common marmosets, and baboons. Altogether, NHP species-based studies highlighted the heterogeneous spectrum of SARS-CoV-2 infection [238,239]. (1) Rhesus and cynomolgus macaques exhibited high levels of viral replication in the upper and lower respiratory tracts, as well as the digestive and urinary tracts [173,174]. AGMs have lower respiratory tract viral titers compared to macaques [181,240], and common marmosets exhibited only low transient levels of viral RNA in nasopharyngeal swab samples [239]. (2) NHP exhibit absent-to-moderate clinical signs characterized by inconstant and benign fever for marmosets and an altered general status for macaques and AGMs [238]. Common marmoset low viral susceptibility and almost absent clinical manifestations make this species less attractive. In contrast, macaques and AGMs experienced weight loss, fever and diffuse alveolar damage, associated with a high permissivity to SARS-CoV-2 [241]. Similarly, baboons had higher viral titers and longer viremia, as well as more pronounced lung pathology relative to SARS-CoV-2 infected macaques [242]. Because SARS-CoV-2 infected baboons experience severe COVID-19 symptoms and breed throughout the year with high fecundity, they should be examined as a supplemental model for biomedical research. As observed in the hamster model, older baboon individuals were more affected by viral infection, having higher viral loads in lung tissues for a longer period time, and an increased number of histopathological changes [3,243]. Aged monkeys could thus be suggested to mimic the most severe form of COVID-19. Interestingly, macaques and AGMs also exhibited virus-specific antibodies with neutralizing activity and T cell responses, making them remarkable models for the study of immune responses after viral challenge [181,244,245]. Rhesus macaque sera samples presented increasingly high levels of neutralizing antibodies 5- and 14-days post-rechallenge, acting as protection against re-infection [175].

NHP models have been mainly useful for therapeutic agents and vaccine candidate testing. Regarding drug assays, remdesivir has been evaluated on rhesus macaques, showing reduced pulmonary infiltrates and reduced viral titers on bronchioalveolar lavage only 12 h after the first dose [176]. Lung tissue autopsies further revealed lower lung viral loads and decreased lung damage in the remdesivir-treated group compared to the control group. HCQ antiviral activity against SARS-CoV-2 infection was also monitored in the cynomolgus model, either alone or in combination with azithromycin [93]. All treated NHP had viral RNA load kinetics similar to untreated NHP at the tracheal, nasopharyngeal, and bronchioalveolar level, and HCQ did not enhance viral clearance duration. mRNA-1273 and BNT162b2 demonstrated a protective effect for both lower and upper airways, as viral subgenomic RNA in bronchioalveolar fluids and in nasal swabs were significantly lower compared to the control group after 2 and 3 days post-inoculation, respectively [246,247]. None of the animals experienced additional weight or temperature change or altered oxygen saturation rates compared to the control group. Regarding conventional vaccines, ChAdOx1-nCoV-19 assays showed much lower levels of the virus in the bronchioalveolar lavage fluid and lower respiratory tract tissues for the vaccinated animals compared to control [248]. In addition, the adenovirus (Ad26) vaccine entered clinical trials after having demonstrated in rhesus macaques that a preventive single administration of the encoding S protein vaccine led to no detectable virus in bronchioalveolar lavage samples and nasal swabs [249].

Although inevitable during the COVID-19 pandemic, NHP represent less than 5% of all animal research [250], as working with these animals constitutes technical and financial issues as well as ethical challenges. Moreover, different challenge dosages and routes of infection in the NHP model may contribute to highly significant alterations in the level and duration of viral replication observed, requiring standardized protocols.

Nevertheless, animal models clarified the positioning of drugs/vaccines during the COVID-19 pandemic, giving precise ideas of their antiviral potency. Finally, constant evaluation and correction of the most appropriate model used during pandemic period is crucial. Selected models should indeed be utilized in the best possible manner in accordance with assumptions made and the latest scientific updates.

## 4. Conclusions & Discussion

The COVID-19 global health crisis rapidly reminded us how great the latency period between the development of adapted state-of-the-art models versus the need to promptly understand disease pathophysiology and to evaluate therapeutic candidates could be. Initially, the lack of consensus regarding study design and experimental readouts sometimes led to unreliable data and general confusion. To date, many different preclinical models are accessible for the investigation of SARS-CoV-2 infection. The in vitro models are mandatory for the understanding of the virus life cycle and compound high-throughput screening. They have the advantage of being quite simple and highly informative in considering virus-host interactions and compound mechanisms of action. Animal models have value in the study of virus transmission, tropism, and evaluation of systemic effects of the infection. These complex models are crucial to assess drug and vaccine efficacy before moving onto clinical trials. A lesson to be drawn from the SARS-CoV-2 outbreak is that none of the models tested so far completely reflects human COVID-19. A more precise examination of the SARS-CoV-2 life cycle and pathophysiology via the diversification of assays will improve the comprehension of virus features and save precious time. The advent of 3D cell-based assays coupled with new emergent engineered technologies is surfacing as the new potent modeling strategy for biomedical research. These human-centered approaches recapitulate cellular heterogeneity, the extracellular environment, and more importantly are exploitable over long periods. Combined-3D models represent quite well the human biology complexity and also overcome animal use limitations, being applicable on a larger scale and therefore facilitating clinical translation. Following such an emergency period, it is of critical importance to pursue modeling enhancement and to recall the relevance of each respective model, with their advantages and limitations.

## Figures and Tables

**Table 1 viruses-14-01507-t001:** Cell-based approaches: key points, benefits, and limitations.

Type	Name (Origin)		Investigation	Benefits	Drawbacks	References
**Immortalized cells**	Vero E6 (African green monkey kidney epithelial cells)	WT	Study of infection mechanism and viral isolation In vitro replication and amplification of viral particles Pharmacological screening	Easy to cultivate High replication rate ACE2 expression CPE Clinical strain isolation and production Viral vectors and vaccines production	No TMPRSS2 receptor expression Non-human cells No direct target cell (kidney cells)	[4,25,32,33]
		TMPRSS2-cells		Easy to cultivate Higher replication rate compared to WT ACE2 and TMPRSS2 expression CPE Clinical strain isolation and production	Non-human cells Not direct virus target (kidney cells)	[30,32,34]
	A549 (Human lung carcinoma)		Study of infection mechanism Comparing tropism, replication kinetics, and cell damage profiling of SARS-CoV-2	Easy to cultivate Moderate replication rate Basal expression level of ACE2 and no TMPRSS2 expression hACE2 and hACE2-TMPRSS2 cells commercially available Main target cell (lung cells)	No TMPRSS2 receptor expression No CPE Minimal viral replication in WT cells without the addition of exogenous proteases	[5,30,33,34,35,36]
	Caco-2 (Human colorectal adenocarcinoma)		Study of infection mechanism Pharmacological screening	Easy to cultivate High replication rate ACE2 and TMPRSS2 expression Main target cell (colon cells)	No CPE Minimal viral replication	[37,38,39]
	Calu-3 (Human non-small lung adenocarcinoma)		Study of infection mechanism Pharmacological screening	Easy to cultivate High replication rate ACE2 and TMPRSS2 expression Main target cell (lung cells)	No CPE Laborious cell culture (time-consuming, low replicative rate) Minimal viral replication Differentiation stage	[30,31,32,35,40,41]
	HEK293T (Human embryonic kidney cells)		Study of infection mechanism Pharmacological screening	Easy to cultivate Modest replication rate ACE2 and TMPRSS2 expression	No CPE Minimal viral replication Not direct virus target (kidney cells)	[29,42,43,44,45,46,47,48]
	Huh7 (Human hepatocellular carcinoma)		Study of infection mechanism and viral isolation Pharmacological screening Study of tropism, replication kinetics and cell damage profiling of SARS-CoV-2	Easy to cultivate High replication rate ACE2 and TMPRSS2 expression	No CPE Low titer of infectious SARS-CoV-2 Not direct virus target (liver cells) No ACE2 expression	[35,42,49,50,51]
	Beas-2B (Human lung, bronchus)		Study of infection mechanism	Easy to cultivate Moderate replication rate CPE hACE2 cells production possible	Controversial ACE2 receptor expression	[52,53]
**Primary cells**	HAE/HAE-ALI (Human airway epithelium)		Study of infection mechanism Model closest to human lungs Diagnosis of SARS-CoV-2 infection and viral isolation Cytokine response profiling and sensitivity to interferons Pharmacological screening	More complex culture and short lifetime ACE2 and TMPRSS2 expression CPE Understanding pro-inflammatory responses of proximal and distal lung epithelium Study of immune signature	Uncontrolled region-to-region and donor-to-donor variation Limited availability of resources and expensive Not possible to study viral evolution	[54,55,56,57,58]
**Organoids**	Human bronchial/lung organoid		Study of infection mechanism Pharmacological screening Mapping of genes associated with SARS-CoV-2 infection Understanding the tissue tropism of SARS-CoV-2	More complex culture High titers of infectious viral particles depending on target organ ACE2 and TMPRSS2 expression depending on studied organ Permissive to SARS-CoV-2 infection Contain a full range of differentiated cell types	Lack of air-liquid interface, vasculature, and immune cells Absence of relevant mechanical cues (air flow, vascular flow) Thick ECM gel complicates permeability and drug studies Not fully recapitulate immune cell population	[59,60,61,62,63,64,65]
	Human colonic/intestinal organoid					
**Organ-on-a-chip**	Primary lung/intestinal epithelium-on-a-chip		Study of infection mechanism Pharmacological screening Mapping of genes associated with SARS-CoV-2 infection Understanding the tissue tropism of SARS-CoV-2 Immune response	Complex culture High titers of infectious viral particles ACE2 and TMPRSS2 expression depending on studied organ Tissue-tissue interfaces Organ-level physical microenvironments Relevant mechanical cues, and fluid flow Closest model to mimic human physiopathology	Relatively low throughput Reproducibility issues	[64,66,67,68]

ACE2: angiotensin-converting enzyme 2, ALI: airway-liquid interface, CPE: cytopathic effect, ECM: extracellular matrix, HAE: human airway epithelium, TMPRSS2: transmembrane protease serine 2, WT: wild type.

**Table 2 viruses-14-01507-t002:** Animal models: key points, benefits, and limitations in COVID-19 investigation.

Type	Model		Key Points	Benefits	Drawbacks	References
**Small animals**	Mice h ACE2	Mouse Ace2 promoter	○ High susceptibility to SARS-CoV-2 infection ○ Experience mild symptoms of COVID-19 patients	**COVID-19:** ○ Absence-to-lethal symptoms: pneumonia, weight loss, reduce activity, increased respiration, cytokine storm ○ Screening of antiviral drugs and vaccines ○ Different susceptibility based on gender	**Model:** ○ Rapid metabolism **COVID-19:** ○ Do not develop severe disease ○ Limited availability during pandemic ○ S protein do not favorably interact with mice ACE2 receptor, must develop transgenic models ○ Potential mouse-to-mouse variation in hACE2 expression and tissue distribution ○ Lethality after brain dissemination	[154]
		Mouse Hfh4 promoter	○ Valuable model for vaccines and drugs assays			[155,156]
		Mouse K18 promoter	○ Investigation of lung ○ Immune and antiviral-based countermeasures			[156,157,158]
		Adenoviral vector	○ Viable model (no brain migration) ○ Allow not permanent genetic mutation ○ Artificial ACE2 expression in non-relevant cell types			[156,159,160]
		CRISPR–Cas9 knock-in	○ Artificial ACE2 expression in relevant cell types ○ High levels of hACE2 expression in lung			[156,161]
	Golden Syrian hamsters		○ Develop severe pneumonia similar to COVID-19 patients ○ Efficiently transmitted from inoculated hamsters to naive hamsters by direct contact and via aerosols ○ SARS-CoV-2-infected hamsters can develop neutralizing antibodies, protecting them from reinfection ○ Develop immunity against reinfection	**COVID-19:** ○ ACE2 expression similar to human ○ Susceptible to SARS-CoV-2 infection ○ Mild-to-moderate symptoms: weight loss, hunched posture, lethargy, respiratory distress ○ Sex and age susceptibility ○ Viral transmission, antiviral drugs and vaccines exploration	**Model:** ○ Do not reflect human pharmacology ○ Fast metabolism (no metabolized drugs test) ○ Cannot be used for long-term pathogenesis due to rapid metabolism **COVID-19:** ○ No severe symptoms ○ Mild weight loss only making hard to test therapies	[162,163,164]
	Roborovski dwarf hamster		○ Develop rapid and consistent on-set of fulminant clinical disease (). ○ Highly susceptible COVID-19 model, having stable and fulminant clinical signs similar to changes described in patients who died from SARS-CoV-2 infection. Such a severe form of COVID-19 has not been reproduced in any experimentally infected animal, positioning this model as a valuable model for screening therapeutics and vaccine candidates in highly susceptible individuals.	**COVID-19:** ○ ACE2 expression similar to human ○ Susceptible to SARS-CoV-2 infection ○ Absence-to-lethal symptoms: weight loss, fall in body temperature, severe acute diffuse alveolar damage, hyaline microthrombi, severe fulminant pneumonia ○ Viral transmission, antiviral drugs and vaccines exploration ○ Relevant central nervous system infection is not observed	**Model:** ○ Do not reflect human pharmacology ○ Fast metabolism (no metabolized drugs test) **COVID-19:** ○ Lethality due to fatal lung pathology	[165]
	Ferrets		○ Effectively transmit the virus by direct or indirect contact ○ Experience mild symptoms of COVID-19 patients	**Model:** ○ Commonly used small animal model ○ Similarities to human respiratory tract ○ Comparable immune system to human **COVID-19:** ○ High susceptibility to SARS-CoV-2 infection ○ ACE2 sequence similar to human ○ Absence-to-moderate symptoms: wheezing, lethargy, sneezing, high temperature ○ Disease transmission, antiviral drugs, immunotherapies, and vaccines exploration	**Model:** ○ Expensive small model due to handling procedures and needs ○ Long-living model ○ May not reflect human pharmacokinetics **COVID-19:** ○ Low viral titer in lungs ○ Not the same severity compared to human	[146,166,167,168,169]
**Large animals**	Cynomolgus macaques		○ Effective virus transmission to other animals ○ Development of lung disease ○ Early peak of virus resembles asymptomatic patients ○ Useful for the evaluation of vaccines, immunotherapies, and antiviral drugs	**Model:** ○ Appreciable size ○ Good longevity ○ Similar pharmacokinetics to humans ○ Innate and adaptive immunity, and physiology exploration **COVID-19:** ○ Similar infection to human ○ ACE2 sequence similar to human ○ Vaccines, immunotherapies, and antiviral drugs exploration	**Model:** ○ Difficult to handle ○ Ethical considerations ○ Small population size **COVID-19:** ○ Limited clinical signs developed	[3,170,171,172]
	Rhesus macaques		○ Exhibits high ACE2-spike activity and is susceptible to SARS-CoV-2 infection ○ SARS-CoV-2 infection induced protective immunity against subsequent reinfection ○ DNA vaccine encoding full-length S protein protected them from SARS-CoV-2 infection ChAdOx1 nCoV-19 vaccine prevented SARS-CoV-2 pneumonia ○ Inactivated SARS-CoV-2 virus vaccine (PiCoVacc) induced SARS-CoV-2-specific neutralizing antibodies ○ Single dose of Ad26 vector-based vaccine protected against SARS-CoV-2 infection			[173,174,175,176,177,178,179,180]
	African green monkeys		○ High level of SARS-CoV-2 replication and pronounced respiratory tract infection ○ Heterologous response along with the ability to collect tissues and longitudinal samples permits a detailed study of pathogenesis and immunity to COVID-19 ○ Developed more severe lung pathology ○ Young healthy AGMs may represent subclinical or mild human disease ○ Tissue tropism and transmission studies			[181,182,183]
	Marmosets		○ Quite new model in research field ○ Not much investigated yet ○ Relatively resistant to SARS-CoV-2 infection			[184,185,186]

WT: wild type, ACE2: angiotensin-converting enzyme 2.

## Data Availability

Not applicable.

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
