# Peer review of "Cell and Animal Models for SARS-CoV-2 Research"

_viruses, 2022, doi:10.3390/v14071507_

Round 1

Reviewer 1 Report

In this review article by Eloïne Bestion and colleagues, the authors comprehensively summarize available cell culture systems and animal models that are (more or less) commonly used to study various aspects of SARS-CoV-2 infection. Although there are numerous reviews that cover the content of this article already, it provides some new aspects and angles that make it worthwhile publishing. I think that it will be suitable for publication following minor edits and some ethical concerns that the authors have to address.

Dr. Halfon provided a company-email (see line 5f), yet you state that you have no competing interests to declare. Please double check if you need to disclose anything according to the journal guidelines!

Author contributions: As per the International Committee of Medical Journal Editors, authorship has to be based on more than just manuscript proofreading, see https://www.icmje.org/recommendations/browse/roles-and-responsibilities/defining-the-role-of-authors-and-contributors.html. I recommend to discuss this issue with all authors and come up with an acceptable solution. As a side note: is “equal contribution” really justified if the contribution actually differed substantially (see line 9/author contributions)?

Title: I would suggest changing the title, also as pathogenesis (in a strict sense) can only be studied in animal models. Agreed? ...and did you see this paper, that has a pretty similar title? https://doi.org/10.1242/dmm.046581 - another reason to revise your title.

Line 24f: please update these numbers. Your reference #2 says that you accessed the WHO homepage on 3 June 2021. ...same is true for reference #4, #11, #37, #67, #90, #126, all these pages were accessed years ago for some reason. 

Line 48: “primordial”?

Line 54: There’s a word missing between “preclinical” and “can”.

Line 56f: You write that “many COVID-19 trials have resulted in disappointing outcomes” but are citing only a single study that shows no effects of drug treatment (reference 14). Please revise or add more references.

The sentence starting in line 81 is incomplete. “In contrast” to Veros that do not express TMPRSS2 as stated in Table 1?

Table 1: I suggest you change all instances of “cell target” to “target cell”. And, is the OOC-approach really “cost and time effective”? It’s the opposite, correct?

Line 100: High-throughput screening of what? Please revise.

Why did you choose to include the information on CQ/HCQ, starting from line 105? If you want to give an example on the importance of the appropriate cell model and highlight the different aspects that researchers have to take care of during drug testing in vitro, I would suggest that you introduce this paragraph with one or two sentences that explain the purpose of this text (lines 105-115).

Line 121: You might want to provide some additional information in these common mutations.

Line 141f: The Pruijssers et al paper has been published in Cell Rep., I suggest that you check all bioRxiv-references (and those that were posted on other preprint servers) if they are actually already published in a peer-reviewed journal and add these if available.

Line 280f: Do you want to mention hamsters in this section, as they also are held as pet animals?

The dwarf hamsters are missing in Tab. 2.

Line 400f: Maybe provide some more examples of vaccine trials and/or therapeutic antibodies in Syrian hamsters?

Line 416: Has this model been used to test vaccines and/or therapeutics already? Maybe provide some examples!?

Suitable references are missing in the sentences starting in line 58, 60, 154 – as well as for sentences ending in lines 245 and 407f.

There’s something wrong with the page numbers.

Brackets are used in a weird way, as “[text)” instead of “(text)” throughout the manuscript.

Finally, the manuscript would benefit from a thorough proofread to reduce language and formatting/style issues. However, maybe small mistakes will be taken care of during copyediting.

Author Response

We really thank the reviewer for the time and effort spent in the review of our manuscript, and for finding our study interesting.

Please see the attachment if easier for you.

  • Halfon provided a company-email (see line 5f), yet you state that you have no competing interests to declare. Please double check if you need to disclose anything according to the journal guidelines!

Thank you for this comment. We have corrected both the affiliations and disclosures on lines 570.

  • Author contributions: As per the International Committee of Medical Journal Editors, authorship has to be based on more than just manuscript proofreading, see https://www.icmje.org/recommendations/browse/roles-and-responsibilities/defining-the-role-of-authors-and-contributors.html. I recommend to discuss this issue with all authors and come up with an acceptable solution. As a side note: is “equal contribution” really justified if the contribution actually differed substantially (see line 9/author contributions)

Thank you for your commentary. We have accordingly adjusted each author contribution providing more details information about the relative work. The adjustments are visible lines 566-569.

  • Title: I would suggest changing the title, also as pathogenesis (in a strict sense) can only be studied in animal models. Agreed? ...and did you see this paper that has a pretty similar title? https://doi.org/10.1242/dmm.046581 - another reason to revise your title.

Accordingly to your comment, we changed the titled « Positioning cell and animal models for studying SARS-CoV-2 pathogenesis » for the new title « Cell and animal models for SARS-CoV-2 research ». We hope this new title will be suitable.

  • Line 24f: please update these numbers. Your reference #2 says that you accessed the WHO homepage on 3 June 2021. ...same is true for reference #4, #11, #37, #67, #90, #126, all these pages were accessed years ago for some reason. 

We do apologize for the wrong consultation dates regarding theses references. We have updated the figures and the information without updating the last consultation date of these citation links. The mistake has been corrected as visible in blue in reference list.

  • Line 48: “primordial”?

The sentence « In vitro models are primordial for studying virus biology under highly con-trolled conditions. » has been replaced by « In vitro models are compulsory for studying virus biology under highly con-trolled conditions. » using more appropriate word.

  • Line 54: There’s a word missing between “preclinical” and “can”.

Thank you for this comment. We added the missing word « exploration ».

  • Line 56f: You write that “many COVID-19 trials have resulted in disappointing outcomes” but are citing only a single study that shows no effects of drug treatment (reference 14). Please revise or add more references.

This comment led us to correctly reformulate the concerned sentence in order to improve the text. The sentence has been amended by the new one: “To date, it has been reported disappointing outcomes regarding COVID-19 trials, failing to benefit patients [11, 14]”.

  • The sentence starting in line 81 is incomplete. “In contrast” to Veros that do not express TMPRSS2 as stated in Table 1?

Accordingly to your comment, we specified the sentence line 82.

  • Table 1: I suggest you change all instances of “cell target” to “target cell”. And, is the OOC-approach really “cost and time effective”? It’s the opposite, correct?

Thank you for the comment. We removed this quite unapproached comment.

  • Line 100: High-throughput screening of what? Please revise.

The sentence « 2D cell-based models are relevant in understanding the physiopathology and are suitable for high-throughput screening against SARS-CoV-2. » has been corrected by « 2D cell-based models are relevant in understanding the physiopathology and are suitable for high-throughput screening of drugs against SARS-CoV-2. ».

  • Why did you choose to include the information on CQ/HCQ, starting from line 105? If you want to give an example on the importance of the appropriate cell model and highlight the different aspects that researchers have to take care of during drug testing in vitro, I would suggest that you introduce this paragraph with one or two sentences that explain the purpose of this text (lines 105-115).

Here is raised a very good point. The choice of CQ/HCQ drugs to highlight the importance of the appropriate cell model and the different aspects that researchers have to take care of during drug testing in vitro is now introduced with two sentences lines 107-110.

  • Line 121: You might want to provide some additional information in these common mutations.

For sure, we therefore directly specified ones of the most common mutations found into the available cell lines used to study coronaviruses and added references.

  • Line 141f: The Pruijssers et al paper has been published in Cell Rep., I suggest that you check all bioRxiv-references (and those that were posted on other preprint servers) if they are actually already published in a peer-reviewed journal and add these if available.

Thank you for this nice comment. We have actualized publications citations when they have been published elsewhere than preprint servers.

  • Line 280f: Do you want to mention hamsters in this section, as they also are held as pet animals?

We thank the reviewer for this comment. We decided to mention hamsters only in “Animal models for SARS-CoV-2 research” since litterature regarding domestic hamster-to-human transmission is really poor. There is an interesting preprint on that subject available on The Lancet website (http://dx.doi.org/10.2139/ssrn.4017393), still waiting for peer-review.

  • The dwarf hamsters are missing in Tab. 2.

We accordingly added the Roborovski dwarf hamster model into the Tab.2 to be more exhaustive.

  • Line 400f: Maybe provide some more examples of vaccine trials and/or therapeutic antibodies in Syrian hamsters?

Thank you for this comment. We consequently added two major recent papers dealing with the investigation of new vaccines efficacy in Syrian hamster model lines 408-415.

  • Line 416: Has this model been used to test vaccines and/or therapeutics already? Maybe provide some examples!?

Roborovski dwarf hamster has not been extensively used to test vaccines and/or therapeutics compared to its cousin Syrian hamster. Meanwhile, we added the last preprint published mentioning the interest of molnupiravir lines 429-432 after having been tested in this animal model.

  • Suitable references are missing in the sentences starting in line 58, 60, 154 – as well as for sentences ending in lines 245 and 407f.

Of course, we therefore added references for each concerned sentences lines 58, 61, 156, 251 and 421, respectively.

  • There’s something wrong with the page numbers.

We appreciate the very conscientious revision. Indeed, something was wrong due to the insertion of page with landscape format. Every pages (expect the first one), have now the same page header.

  • Brackets are used in a weird way, as “[text)” instead of “(text)” throughout the manuscript.

I apologize for this repetitive typing error. All non-appropriate [ have been replaced by (.

  • Finally, the manuscript would benefit from a thorough proofread to reduce language and formatting/style issues. However, maybe small mistakes will be taken care of during copyediting.

We thank you for this recommendation. We improved formatting/style issues the best as we can and language issues have normally been corrected before as English of the paper has been revised by a professional.

Reviewer 2 Report

Bestion et al. provide a  review encompassing cell-based approaches and preclinical animal models of SARS-CoV-2 infection developed since the beginning of the COVID19 pandemic and discussed their advantages and drawbacks and their relevance to anti-viral drug evaluation. No single model is ideal for perfectly recapitulating COVID-19 in humans after SARS-Cov2 infection. The authors discussed emerging techniques applied to SARS-CoV-2 research as well as the importance of the identification of animal reservoirs to guarantee effective disease control. Lastly, the authors described the necessity of compiling models to correctly summarize SARS-CoV-2 pathophysiology and to de-risk clinical trial failure.

The manuscript is easy to read and well explains gaps between clinical disease manifestations and preclinical experimental conditions using cells and animal models in understanding the pathogenesis of SARS-CoV 2 infection and COVID19, identification of therapeutic targets, and screening platform of anti-viral reagents.  I have a few minor concerns as indicated below.

Check the references. A few citations are incorrect.

Line 54, the sentence is incomplete; ‘Extensive preclinical … (need something here) can identify lead candidates, …’

Line 52, who are ‘they’? if the author meant in vivo models, revise it to ‘these models’

Line 231, the primary author of citation no. 94 is ‘Si et al.’ not Ingber et al.

Line 262, correct typo and add a space ‘White-tailed deer express’

Page 3, Table 2, citation 132 a paper using a mouse model not NHP models

Line 160, indicate citation ‘…. with ACE2 inhibition [59]’

Line 423, combine the references [161,162],[163] to [161-163]

Author Response

We really thank the reviewer for the time and effort spent in the review of our manuscript, and for finding our study interesting.

Please see the attachment if easier for your revision.

  • Check the references. A few citations are incorrect.

Thank you for your comment. References have been revised in the latest version of the manuscript accordingly to your remark.

  • Line 54, the sentence is incomplete; ‘Extensive preclinical … (need something here) can identify lead candidates, …’

Thank you for this comment. We added the missing word « exploration ».

  • Line 52, who are ‘they’? if the author meant in vivo models, revise it to ‘these models’

We appreciate your effort to make the text more intelligible. We accordingly made the modification.

  • Line 231, the primary author of citation no. 94 is ‘Si et al.’ not Ingber et al.

Thank you for taking the time to check this. We have corrected the mistake.

  • Line 262, correct typo and add a space ‘White-tailed deer express’

We corrected typo line 268.

  • Page 3, Table 2, citation 132 a paper using a mouse model not NHP models

Sure, the citation was not correct at this part of the Table. We corrected the error.

  • Line 160, indicate citation ‘…. with ACE2 inhibition [59]’

Thank you. The citation has been added line 166 in order to improve the understanding.

  • Line 423, combine the references [161,162],[163] to [161-163]

We apologize for the typo mistake. It is now corrected.
